# Test and Numerical Model of Curved Steel–Concrete Composite Box Beams under Positive Moments

**DOI:** 10.3390/ma14112978

**Published:** 2021-05-31

**Authors:** Zhi-Min Liu, Xue-Jin Huo, Guang-Ming Wang, Wen-Yu Ji

**Affiliations:** 1School of Civil Engineering, Beijing Jiaotong University, Beijing 100044, China; zhmliu@bjtu.edu.cn (Z.-M.L.); 14115279@bjtu.edu.cn (G.-M.W.); wyji@bjtu.edu.cn (W.-Y.J.); 2China Railway Major Bridge Reconnaissance & Design Institute Co., Ltd., Wuhan 430056, China

**Keywords:** curved steel–concrete composite box beams, coupled bending and torsion, experimental research, elaborate finite element model, elastoplastic behavior

## Abstract

Compared with straight steel–concrete composite beams, curved composite beams exhibit more complicated mechanical behaviors under combined bending and torsion coupling. There are much fewer experimental studies on curved composite beams than those of straight composite beams. This study aimed to investigate the combined bending and torsion behavior of curved composite beams. This paper presents static loading tests of the full elastoplastic process of three curved composite box beams with various central angles and shear connection degrees. The test results showed that the specimens exhibited notable bending and torsion coupling force characteristics under static loading. The curvature and interface shear connection degree significantly affected the force behavior of the curved composite box beams. The specimens with weak shear connection degrees showed obvious interfacial longitudinal slip and transverse slip. Constraint distortion and torsion behavior caused the strain of the inner side of the structure to be higher than the strain of the outer side. The strain of the steel beam webs was approximately linear. In addition, fine finite element models of three curved composite box beams were established. The correctness and applicability of the finite element models were verified by comparing the test results and numerical calculation results for the load–displacement curve, load–rotational angle curve, load–interface slip curve, and cross-sectional strain distribution. Finite element modeling can be used as a reliable numerical tool for the large-scale parameter analysis of the elastic–plastic mechanical behavior of curved composite box beams.

## 1. Introduction

Steel–concrete composite beams have been widely used in the construction of engineering structures due to their light weight, strong spanning ability, high bearing capacity, and association with quick construction. However, unlike that of straight composite beams, the mass center of curved composite beams is not at the connections of the supporting points on both ends, so typical bending–torsion coupling stress characteristics can be found in curved composite beams. The deflection, torsion angle, and even interface slip between the steel beam and the concrete slab of a curved composite beam, owing to bending–torsion coupling, are more obvious than those of a straight composite beam. In addition, the kinds of torsion include free torsion and constrained torsion, and the latter more commonly occurs in structures with complex boundary conditions. Constrained torsion leads to a more complex and irregular strain distribution in the structure. Overstressing may occur when design schemes suitable for linear composite beams are applied to curved composite beams.

Generally, model tests are the most direct, effective, and accurate method for studying structural force behavior. There have been a large number of test studies on straight composite beams and relatively few studies on curved composite beams. Colville [1] conducted a static test study on four curved composite beams and proposed a simplified design method for interfacial shear connectors. Brennan and Mandel [2] conducted a static test study on a large-scale two-span continuous curved composite I-beam and studied the displacement and stress distribution under normal usage load. The Federal Highway Administration (FHWA) of the United States Department of Transportation [3] has carried out a series of experimental studies on the force behavior of curved beams, including a static test of a full-scale curved composite box I-beam that focuses on monitoring the displacement and internal force change of the structure under dead load, live load, and ultimate load. Thevendran et al. [4] conducted a static test study on five curved composite I-beams with different curvatures under concentrated load in the middle of the span, and the test results showed that the ultimate bearing capacity decreased with the ratio of the span to the radius. Krzmarzick and Hajjar (2006) [5] studied the trends of the displacement and stress distribution of a curved composite I-beam bridge through an actual bridge load test. Tan and Uy (2009) [6] carried out a static test study on eight curved composite I-beams. The test parameters included the ratio of the span to the radius and shear connection degree. The test results showed that curved composite beams enduring the bending moment could increase the torsional bearing capacity, but the flexural bearing capacity of the structure under torque was not affected. Zhang et al. (2012) [7] conducted static tests on six curved composite box beams under the combined action of bending and torsional loads. The test parameters were the ratio of the torque to the moment and the number of diaphragms. The test results showed that the failure modes of the specimens included bending failure and bending–torsion coupled failure; this failure mode increased the flexural bearing capacity of the structure to a certain extent when the structure was under torque, and significant longitudinal and transverse slip existed at the interface between the steel beam and the concrete slab. Through the static tests of two composite beams, Zhang concentrated on studying the effect of a large curvature, such as a width-to-radius ratio greater than 1/10, on the force behavior of the structure. Zhu [8] conducted a static loading test on a large-curvature-curve composite box beam with a positive bending moment under normal usage load. In 2020, Zhu [9] carried out a static loading test on a large-curvature-curve composite box beam with a negative bending moment under ultimate load, and the test results showed that, due to the large curvature, the stress on the inner side of the structural slabs was significantly greater than that on the outer side.

Compared to actual model tests, numerical models are not restricted by the production cost, geometric size, or boundary conditions. Under the premise that the accuracy of the numerical model has been verified, the structural parameters and boundary conditions can be changed through the numerical model to systematically analyze the force behavior of the structure. Numerical models of curved composite beams include frame models and fine models. A frame model is highly efficient, and a fine model is highly accurate. With the development of numerical computing capabilities, the efficiency of fine models in modeling and calculation has been greatly improved. Therefore, fine models can be used to analyze the parameters of curved composite beams at present. Ei-Tawil et al. [10] established a frame model and numerical model of a curved composite box beam. With the aid of parameter analysis, the influence of the curvature, cross-sectional geometric parameters, and the number of beam spans on the structural force behavior was studied. The results showed that the warping stress contributes to the stress level to a certain degree. Sennah et al. and Samman et al. [11,12,13,14] adopted fine models and conducted many studies on the force behavior of curved composite box beams under normal usage and the lateral load distribution coefficient among multibox beams, and they proposed a simplified calculation method for the lateral load distribution. Daniel et al. [15] studied the dynamic behavior of a curved composite I-beam under seismic loading through a fine model. The numerical model test parameters included the curvature, I-beam distance, and the structural form of the diaphragm. The research results showed that the curvature had a significant impact on the seismic responses of curved composite I-beams. Additionally, the curvature, I-beam distance, and diaphragm structure had a more significant influence on the inner side of the beam than on the outer side. Nie and Zhu [16] systematically studied the modeling strategies of fine models and frame models of curved composite box beams. A fine model and a frame model of an actual curved composite box beam bridge were established, and the accuracy and applicability of the established models were verified. Lin and Yoda [17] established a fine model based on a static load test of a straight composite beam with a negative bending moment; they extended it to the analysis of the elastoplastic mechanical behavior of a curved composite beam with a negative bending moment, studied the bearing capacities of structures and the strain distribution trends with changing curvature, and proposed a calculation formula suitable for the bearing capacity considering the coupling effect of bending and torsion. Fatemi et al. [18] established a fine model of curved composite box beams, studied the influence of the curvature, span, and number of lanes and box beams on the force behavior of the structure, and determined the structural form and reasonable space of the lateral support of structures. Zhu et al. [19] and Wang et al. [20] established a fine model of a curved composite box beam; this model was used as a benchmark for the frame model of a curved composite box beam considering constrained torsion, distortion, and bidirectional interface slip. Sucharda et al. [21] and Valikhani et al. [22] presented the possibilities of material models of concrete, approaches to the choice of parameters, or taking into account the uncertainties in the calculation or stochastic character of concrete. These strategies can be introduced to the FE simulation of curved composite beams.

In contrast to the experimental and numerical studies of straight composite beams, there are relatively few studies on curved composite beams. Additionally, there are relatively few experimental and numerical studies on curved composite box beams compared to studies on curved composite I-beams. In China, in overpasses and expressway ramp bridges, the number of curved composite box beams is greater than that of curved composite I-beams. To further enrich the test and numerical research results of curved composite box beams, this study included static loading tests on three curved steel–concrete composite box beam specimens to investigate their combined bending and torsion behaviors. The parameters of the specimens refer to the central angle and the degree of interfacial shear connection. The load–displacement curves, load–rotational angle curves, load–interface slip curves, and cross-sectional strain distributions of the specimens were reported. After that, the fine finite element models of the three specimens were established, and the test results were compared with the finite element calculation results to establish the accuracy and applicability of the finite element model. As a result, the model can be used as a numerical tool for subsequent parameter analysis.

## 2. Materials and Method

### 2.1. Specimen Design

As per the Chinese code (CMC 2010 [23], 2017 [24]), three curved steel–concrete composite box beams were designed with different central angles and interface shear connection degrees, and the parameters of the three curved composite box beam specimens are shown in Table 1. Each specimen was composed of steel girders and concrete slabs and connected by shear studs. The steel beam consisted of a top flange, bottom flange, steel web, longitudinal and transverse stiffeners, and diaphragms. The span length (arc length between supports) was 6200 mm. The concrete slab was 750 mm in spacing and 50 mm in thickness. The tub steel box girder was 350 mm in width (the distance between the two webs) and 300 mm in height. The bottom flange was 410 mm in width and 12 mm in thickness. The top flange was 100 mm in width and 8 mm in thickness. The steel web was 280 mm in height and 12 mm in thickness. Seven diaphragms were applied at equal intervals along the beam span, and the thickness of the diaphragm was 8 mm. A total of 8 longitudinal steel rebars were arranged in the concrete slab with a spacing of 95 mm, and a total of 82 transverse steel rebars were arranged, each 12 mm in diameter. Samples CCB-1, CCB-2, and CCB-3 had central angles of 45°, 25°, and 45°, respectively. The shear studs were uniformly arranged in 83, 83, and 23 rows, respectively, along the longitudinal direction and connected the steel top flange and the concrete slab. Five shear studs were applied at the top of the end diaphragms of all specimens, and two studs were arranged on the five diaphragms in the middle to connect the diaphragms and the concrete slabs. The shear studs were 13 mm in diameter and 40 mm in height. The basic structure of the test beam is shown in Figure 1. The geometric dimensions and details of the test beam are shown in Figure 2.

The construction process of the test specimens is shown in Figure 3. The steel beams, formwork, reinforcement, and shear studs were manufactured and welded in the factory, and the concrete slab was poured in the factory.

### 2.2. Material Properties

The steel plate had a strength grade of Q345c, and the specimens included two steel plate thicknesses: 8 and 12 mm. The longitudinal and transverse reinforcement had a strength grade of HRB335, and the rebar was 12 mm in diameter. As per the Chinese code (CMC 2010 [23], 2017 [24]), four coupon tests were conducted for steel plates of each thickness and reinforcement, and the average strength and elongation ratio are shown in Table 2.

The strength grade of the concrete was C50. When the concrete slab of each beam was poured, three 150 mm standard cube concrete test specimens of the same material were prepared as per the Chinese code (CMC 2009 [25]). Table 3 gives the mix proportion of C50 concrete coupons. The cube concrete test specimens were poured and cured under the same conditions as the concrete slabs of the specimens, and the material properties of the cube concrete test specimens were tested on the same day of loading. The 150 mm cube compressive strengths of the three cube concrete test specimens are shown in Table 4.

### 2.3. Test Setup

A static failure loading test was conducted for the curved composite box beam specimens at the midspan section through a vertical actuator. The loading device is shown in Figure 4, where Figure 4a,b show the CCB-1 positive bending moment failure, and the loading devices of CCB-2 and CCB-3 were the same as that of CCB-1. In addition, the inner side of the end of each specimen was connected to the steel beam and the ground beam by anchor bolts to prevent the curved composite beam specimen from overturning during the loading process. The loading beam, sensors, and vertical actuator were arranged. The hierarchical loading was used. Before the calculated yield load was reached, force control was applied. After the yield load was reached, the loading was switched to displacement control.

### 2.4. Instrumentation

In this test, the load and strain of the specimens were recorded by a computer data processing system, a displacement meter was used to record the vertical displacement and interface slip of the specimens, and a microscope for crack measurement was used to measure the crack width of the concrete slab. The measuring point arrangement scheme of the CCB-1 specimen is shown in Figure 5, and the measuring arrangement schemes of CCB-2 and CCB-3 were the same as that of the CCB-1 specimen. The midspan section and the 1/4-span section were selected as the control sections, and 16 strain gauges were arranged at each control section to measure the normal strain of the curved beam. The strain gauges were arranged on the top surface of the concrete slab, the longitudinal reinforcement in the concrete slab, and the bottom flange and web side of the steel girder, as shown in Figure 5. Two vertical displacement transducers were arranged in each of the midspan sections and the 1/4-span section to measure the vertical displacement and rotational angle of the curved beam. The two displacement transducers were located directly below the inner and outer webs, as shown in Figure 5c–e. A displacement transducer placed along the longitudinal direction was arranged at the end fulcrum section to measure the longitudinal interface slip of the specimens, as shown in Figure 5a,c. Two displacement transducers placed along the transverse direction were arranged on the midspan section to measure the transverse interface slip of the specimens, as shown in Figure 5a,b.

## 3. Results and discussion

### 3.1. Test Observations

Figure 6a shows the vertical displacement and torsion of CCB-2. The figure indicates that the curved composite box beam showed typical bending–torsion coupling stress characteristics under the vertical load, and the vertical displacement and torsion of the other three specimens were similar. Figure 6b shows the longitudinal slip that occurs at the end of CCB-3. Due to the sparse arrangement of the studs and weak shear connections, CCB-3 presented significant longitudinal slip at the end of the beam, while those of the other specimens were not obvious. The crack distribution of CCB-3 is shown in Figure 7. The crack distributions of CCB-1 and CCB-2 were similar to that of CCB-3. Figure 7a shows that the concrete slab of CCB-3 was compressed during the loading process, and some cracks developed in the transverse direction due to tension in the midspan, whereas more shear diagonal cracks due to torsion occurred on both ends and extended from the end to the middle.

### 3.2. Load–Displacement Curve and Capacity

Figure 8 gives the load–displacement curves of CCB-1–CCB-3. Figure 8a presents the load–displacement curves of the midspan sections of CCB-1–CCB-3, and Figure 8b presents the load–displacement curves of the 1/4-span sections of CCB-1–CCB-3. The vertical displacement of the specimen is equal to the average of the values measured by the displacement sensor directly under the inner and outer webs. With the loading process, the vertical displacement of the specimen increased significantly, and the growth rate slowed with increasing load.

Figure 8 shows that the ultimate bearing capacity of CCB-2 was 11.4% higher than that of CCB-1. The main reason is that the cubic compressive strength of the concrete slab of CCB-2 was higher than that of CCB-1; the secondary reason is that the central angle of CCB-1 was larger, and the bearing torque that weakens the flexural bearing capacity was stronger. The ultimate displacement of CCB-2 was 7.4% larger than that of CCB-1; the ultimate bearing capacity of CCB-1 was 1.9% higher than that of CCB-3, and its ultimate displacement was 1.2% less than that of CCB-3. The shear connection degree had no obvious effects on the ultimate bearing capacity and ultimate displacement of the specimen. The same phenomena were found for the study conducted by Nie and Cai [26]. The initial stiffness of CCB-1 was weaker than that of CCB-2 due to the more significant bending and torsion coupling behavior in CCB-1 and greater vertical displacement; the initial stiffness of CCB-3 was weaker than that of CCB-1, and the interface slips as a weak shear connection reduced the structural rigidity.

Figure 9 gives the load–rotational angle curves of CCB-1–CCB-3. Figure 9a presents the load–rotational angle curves of the midspan sections of CCB-1–CCB-3, and Figure 9b presents the load–rotational angle curves of the 1/4-span sections of CCB-1–CCB-3. The torsion angle of the specimen is equal to the ratio of the difference in the value measured by the displacement sensor directly under the inner and outer webs to the distance between the inner and outer webs. With the loading process, the rotational angle of the specimen increased significantly, and the growth rate slowed with increasing load. The developing trends of and the differences between the load–rotational angle curves of the three specimens were similar to the characteristics of their load–displacement curves.

Figure 10 gives the load–interface slip curves of CCB-1–CCB-3. Figure 10a presents the load–interface slip curves of the 1/4-span section of CCB-1–CCB-3, and Figure 10b represents the load–interface slip curves of the sections on both ends of CCB-1–CCB-3. The transverse slip and vertical slip of the specimen were equal to the values measured by the displacement sensor at the midspan section and support sections on both ends, respectively. With the loading process, the interface slip of the specimen increased significantly, and the growth rate slowed with increasing load. Note that because the shear connection degree directly affects the interface slip, the initial stiffness of CCB-3 was weakest, and the ultimate slip was much larger than those of the other specimens.

The yield load, yield displacement, ultimate bearing capacity, and ultimate displacement of CCB-1–CCB-3 are shown in Table 5. The yield load of a specimen is obtained by the illustrated method, and the corresponding displacement is the yield displacement. As the steel beams or rebars of the three specimens do not break or fail during the loading process, the energy inside the structure is not released, and the load that the structure endures is constantly increasing. Ultimately, the test is stopped due to the loading capacity of the jack. As a result, the peak load of the specimens is their own ultimate bearing capacity, and the corresponding displacement is the ultimate displacement. Table 5 shows that the yield load was directly related to the initial stiffness of the load–displacement curve. The lower the initial stiffness of the specimen, the lower the yield load.

### 3.3. Strain Distribution

Figure 11a,b show the normal strain distribution of the concrete slab of CCB-1 at the midspan section and the 1/4-span section, respectively; Figure 11c,d show the normal strain distribution values of the concrete slab of CCB-2 at the midspan section and the 1/4-span section, respectively; Figure 11e,f show the normal strain distribution values of the concrete slab of CCB-3 at the midspan section and the 1/4-span section, respectively.

As shown in Figure 11, the normal strain of the top slab of CCB-1 was larger than that of the top slab of CCB-2, which indicates that an increase in the central angle improved the overall level of structural strain. Additionally, the normal strain of the top slab of CCB-3 was slightly larger than the normal strain of the top slab of CCB-1, which indicates that the shear connection degree had no significant effect on the strain. The normal strain of the concrete slab at the midspan section and the 1/4-span section of each specimen was distributed unevenly in the transverse direction, and that of the inner side was larger than that of the outer side. This is due to the combined influence of restrained torsion and distortion effects. In particular, CCB-3 showed a very small tensile strain on the outer side of the concrete slab.

Figure 12 shows the normal strain distribution of the steel bottom flange of CCB-1 at the midspan section and the 1/4-span section. The key areas of the midspan sections of CCB-1–CCB-3 reached yield strains at 0.6*P*_u_, 0.8*P*_u_, and 0.6*P*_u_, respectively. The key areas of the 1/4-span sections of CCB-1 and CCB-3 reached yield strains at 0.8*P*_u_ and 0.8*P*_u_, respectively, and the 1/4-span section of CCB-2 was in the elastic stage. The trends of and differences in the strain distributions of the steel bottom flanges of each specimen were basically consistent with those of the concrete slab.

Figure 13 gives the normal strain distributions of the steel webs of CCB-1–CCB-3 at the midspan sections and the 1/4-span sections. The vertical coordinate 0 in the figure represents the position of the interface of the composite beam, and the vertical coordinate 290 mm represents the lower flange slab of the steel beam. The figure shows that the strain distribution of the section was close to a straight line along the direction of the beam height.

Figure 13 shows that the key positions of the inner webs in the midspan sections of CCB-1–CCB-3 reached yield strains of 0.6*P*_u_, 0.8*P*_u_, and 0.6*P*_u_, respectively; the midspan section of the outer web of CCB-2 reached a yield strain of 0.8*P*_u_; and the midspan sections of the outer webs of CCB-1 and CCB-3 were both in the elastic stage. The key positions of the inner webs in the 1/4 sections of CCB-1–CCB-3 reached yield strains of 0.8*P*_u_, 0.8*P*_u_, and 0.8*P*_u_, respectively; the 1/4 section of the outer webs of CCB-1–CCB-3 were all in the elastic stage.

## 4. Finite Element Analysis

### 4.1. Elaborate Finite Element Model

In this paper, the force behaviors of CCB-1–CCB-3 during the whole loading process were simulated in ABAQUS [27]. Figure 14 shows the fine finite element model of CCB-2, and the fine finite element models of the other two specimens were similar. In terms of element selection, SOLID elements were used to simulate the concrete slabs, SHELL elements were used to simulate the steel beams, TRUSS elements were used to simulate the vertical and transverse rebars, CONNECTOR elements were used to simulate the studs, and SOLID elements were used to simulate the loading beams. In terms of the interaction condition settings between the elements, the TRUSS elements of the rebars were imported into the SOLID elements of the concrete slabs by the EMBED command, ignoring the bond–slip effects between the rebar and the concrete. When the CONNECTOR used elements to simulate the studs, vertical displacement between the concrete slab and the steel beam did not occur, which indicates that the vertical stiffness of the CONNECTOR element was infinitely large. Additionally, the vertical load was set on the midspan of the beam, and the model boundary conditions were simply supported. For meshing, the unit size was 25 mm, and the accuracy of the mesh convergence was verified by a mesh test.

The constitutive relations of the materials in the test were as follows:(1)Concrete

For compressed concrete, the yield rule is the von Mises yield surface, the hardening rule is isotropic strengthening, and the flow rule is associated flow. The concrete damage plastic (CDP) model is used. According to the Wang et al. study [28], some critical parameters, including the parameter controlling the projection of the yield surface onto the bias place *K*_c_ = 0.67, the expansive angle related to the flow rule *φ* = 37–42°, and the eccentricity of the plastic potential function *λ* = 0.1, are determined. The uniaxial compression stress–strain relationship recommended by Hognestad et al. [29] is adopted, as shown in Figure 15a. The compressive strength of the concrete cylinder is calculated according to the suggestion of Chen et al. [30]. Equations (1) and (2) give the calculation formula of the curve in Figure 15a.

According to the research of Guo [31], CEB-FIP 1990 [32], and Bazant and Oh [33], the uniaxial tensile stress–strain relationship of concrete is shown in Figure 15a. Equations (3)–(5) give the calculation formula of the curve in Figure 15a.
(1)$$\frac{\sigma_{c}}{f_{c}^{\prime }}=\{\begin{array}{cc} 2\frac{\varepsilon_{c}}{\varepsilon_{0}}-{\left(\frac{\varepsilon_{c}}{\varepsilon_{0}}\right)}^{2} & \varepsilon_{c}\leq \varepsilon_{c0}\\ 1-0.85\left(\frac{\varepsilon_{c}-\varepsilon_{c0}}{\varepsilon_{\mathrm{cu}}-\varepsilon_{c0}}\right) & \;\varepsilon_{c0}<\varepsilon_{c}\leq \varepsilon_{\mathrm{cu}} \end{array}$$
(2)$$f_{c}^{\prime }=\left\{ \begin{array}{ll} 0.8f_{\mathrm{cu}} & f_{\mathrm{cu}}\leq 50\\ f_{\mathrm{cu}}-10 & f_{\mathrm{cu}}>50 \end{array}\right.$$
where $\varepsilon_{c}$ is the strain of concrete; $\varepsilon_{c0}$ is the peak compressive strain of concrete, taken as 0.002; $\varepsilon_{cu}$ is the ultimate compressive strain of concrete, taken as 0.0033; $f_{c}^{\prime }$ is the compressive strength of the concrete cylinder; and $f_{\mathrm{cu}}$ is the compressive strength of the concrete cube, which can be obtained from the results of the material property tests in Table 4.
(3)$$\frac{\sigma_{c}}{f_{\mathrm{ct}}}=\{\begin{array}{cc} \frac{\varepsilon_{c}}{\varepsilon_{\mathrm{ct}0}} & \varepsilon_{c}\leq \varepsilon_{\mathrm{ct}0}\\ \frac{\varepsilon_{\mathrm{ctu}}-\varepsilon_{c}}{\varepsilon_{\mathrm{ctu}}-\varepsilon_{\mathrm{ct}0}} & \varepsilon_{\mathrm{ct}0}<\varepsilon_{c}\leq \varepsilon_{\mathrm{ctu}} \end{array}$$
(4)$$f_{\mathrm{ct}}=\left\{ \begin{array}{ll} 0.26f_{\mathrm{cu}}{}^{2/3} & f_{\mathrm{cu}}\leq 50\\ 0.21f_{\mathrm{cu}}{}^{2/3} & f_{\mathrm{cu}}>50 \end{array}\right.$$
(5)$$\varepsilon_{\mathrm{ctu}}=\frac{2G_{f}}{f_{\mathrm{ct}}l_{e}}$$
where $f_{\mathrm{ct}}$ is the tensile strength of concrete; $\varepsilon_{\mathrm{ct}0}$ is the cracking strain of concrete; $\varepsilon_{\mathrm{ctu}}$ is the ultimate tensile strain of concrete; $G_{f}$ is the fracture energy of concrete calculated according to CEB-FIP1990 [32]; and $l_{e}$ is the characteristic length of elements of concrete calculated according to CEB-FIP1990 [32].(2)Steel plate

For steel plates, the yield is represented by the von Mises yield surface, the hardening is represented by kinematic hardening, and the flow is represented by associated flow. The uniaxial stress–strain relationship of the steel is represented by the typical trilinear model according to the reference [34], as shown in Figure 15b. Equation 6 gives the calculation formula of the curve in Figure 15b.
(6)$$\sigma_{s}=\{\begin{array}{cc} E_{s}\varepsilon_{s} & \varepsilon_{s}\leq \varepsilon_{y}\\ f_{y} & \varepsilon_{y}<\varepsilon_{s}\leq \varepsilon_{h}\\ f_{y}+\frac{f_{u}-f_{y}}{\varepsilon_{\mathrm{su}}-\varepsilon_{h}}\left(\varepsilon_{s}-\varepsilon_{h}\right) & \varepsilon_{s}>\varepsilon_{h} \end{array}$$
where $\varepsilon_{y}$ is the yield strain of the steel; $\varepsilon_{h}$ is the strain when the steel begins to strengthen, taken as 12$\varepsilon_{y}$ according to Han’s research [35]; $\varepsilon_{\mathrm{su}}$ is the ultimate strain of the steel; $f_{y}$ is the yield strength of the steel, which can be obtained from the results of the material property tests in Table 2; and $f_{u}$ is the ultimate strength of the steel.(3)Rebar

For the rebar, the yield is represented by the von Mises yield surface, the hardening is represented by kinematic hardening, and the flow rule is represented by associated flow. The uniaxial stress–strain relationship of the steel is represented by the typical ideal elastoplastic model according to the reference [34], as shown in Figure 15c. Equation (7) gives the calculation formula of the curve in Figure 15c.
(7)$$\sigma_{r}=\left\{ \begin{array}{ll} E_{s}\varepsilon_{r} & \varepsilon_{r}\leq \varepsilon_{\mathrm{ry}}\\ f_{\mathrm{ry}} & \varepsilon_{r}>\varepsilon_{\mathrm{ry}} \end{array}\right.$$
where $\varepsilon_{r}$ is the strain of the rebar; $\sigma_{r}$ is the stress of the rebar; $\varepsilon_{\mathrm{ry}}$ is the yield strain of the rebar; and $f_{\mathrm{ry}}$ is the yield strength of the rebar, which can be obtained from the results of the material property tests in Table 2.(4)Stud

A three-dimensional nonlinear element is used to simulate the studs in ABAQUS, and the relationship of the shear force *V*-slip Δ specified by CEB-FIP2010 [36] is shown in Figure 15d. Equations (8) and (9) give the calculation formula of the curve in Figure 15d.
(8)$$V=V_{u}{\left(1-e^{-\alpha_{2}\Delta }\right)}^{\alpha_{1}}$$
(9)$$V_{u}=\left(5.3-\frac{1.3}{\sqrt{n_{s}}}\right)A_{\mathrm{us}}{f_{c}^{\prime }}^{0.35}f_{\mathrm{ru}}^{0.65}{\left(\frac{E_{c}}{E_{s}}\right)}^{0.4}$$
where $V_{u}$ is the shear bearing capacity of a single stud; $\alpha_{1}$ and $\alpha_{2}$ are the control parameters subjected to force, and according to CEB-FIP2010 [36], $\alpha_{1}$ = 0.75 and $\alpha_{2}$ = 1.1; $n_{s}$ is the number of studs in each group; $A_{\mathrm{us}}$ is the area of a single stud; $E_{c}$ and $E_{s}$ are the elastic moduli of concrete and steel, respectively; and $f_{\mathrm{ru}}$ is the tensile strength of the stud. The kinematic hardening model is used to simulate the hysteretic force behavior of studs.

For the FE model, the material and geometrical nonlinearities are considered. A modified arc-length approach is introduced to the algorithm with the auto-adaptive loading schedule. The boundary condition for the specimen is simply supported. As the concrete damage plastic model cannot simulate the concrete behavior subjected to shear, the abovementioned FE model can only simulate the flexural failure rather than shear failure of the curved composite beams.

### 4.2. Validation and Verification of Finite Element Modeling

The test observations and finite element simulation of the final loading shape of CCB-2 are presented in Figure 16. The curved composite box beam showed typical bending and torsion coupling force characteristics under vertical loading, and the final shapes of the other two specimens were similar. The figure shows that the results of the test observation were in good agreement with those of the finite element simulation.

Figure 17 shows a comparison between the test results and the finite element calculation results of the load–displacement curves of the midspan sections and 1/4-span sections of CCB-1–CCB-3. For the initial stiffness, ultimate bearing capacity, and the developing trend of the whole curve, the results of the test observation were in good agreement with those of the finite element calculations. However, in some partial sections of the load–displacement curves, the test results were slightly smaller than the results of the finite element calculation, which may be due to the welding residual stress decreasing the stiffness in the steel.

Figure 18 shows a comparison between the test results and the finite element calculation results for the load–torsion angle curves of the midspan sections and 1/4-span sections of CCB-1–CCB-3. Figure 19 gives a comparison between the test results and the finite element calculation results of the load–interface slip curves of the midspan sections and 1/4-span sections of CCB-1–CCB-3. Figure 18 and Figure 19 show that the finite element model can simulate the development of the load–torsion angle curve and the load–interface slip curve of the specimen.

Figure 20 shows a comparison between the test results and the finite element calculation results of the normal strain of the concrete slabs of the midspan sections and 1/4-span sections of CCB-1–CCB-3. Figure 21 gives a comparison between the test results and the finite element calculation results for the normal strain of the steel bottom plate of the midspan sections and 1/4-span sections of CCB-1–CCB-3. The finite element model simulates the strain distribution and trends of the structure during the loading process.

In summary, the established fine finite element model can accurately simulate the developing trends of the force behavior of curved composite box beams during the whole loading process, which illustrates that the established fine finite element model can subsequently be used as a powerful numerical calculation tool for large-scale parameter analysis.

### 4.3. Comparison of Experimental, Numerical, and Theoretical Results

The reference [37] gives the theoretical method to calculate the bearing capacity of composite beams. The method does not distinguish the cases of strong and weak shear connection at the interface. Table 6 gives the comparison among the bearing capacities of the three specimens obtained from the experimental, the numerical, and the theoretical results. It can be seen that there are good agreements among the results obtained from the test, the FEM, and the theoretical method.

## 5. Conclusion

In this paper, static loading tests of three curved composite box beams with various central angles and interface shear connection degrees were carried out, and static performance indicators such as vertical displacement, rotational angle, interface slip, and strain of the critical sections were measured. In addition, three-dimensional elaborate finite element models were developed to simulate the test results. Based on the comparison between the finite element models and test results, the following three conclusions can be drawn:

(1) The ultimate bearing capacity and initial stiffness of the curved composite box beam specimens with large central angles were slightly lower than those with small central angles, but within the range of 25° to 45°; the central angle had a limited influence on the ultimate capacity and initial stiffness. Specifically, the ultimate bearing capacity of CCB-2 was 11.4% higher than that of CCB-1 and the ultimate bearing capacity of CCB-1 was 1.9% higher than that of CCB-3. The interface slip of the weak shear connection degree specimen was notably greater than that of the strong shear connection degree specimen. The ultimate interface slip of CCB-1 was twice more than that of CCB-3. The test results showed that the shear connection degree had a significant influence on the interface slip behavior and that the influence of the shear connection degree on the ultimate capacity and initial stiffness was weaker than that on the slip behavior.

(2) The positive strain of the top flange plate and the steel bottom plate of the curved composite box beam specimens was nonuniform along the transverse direction, and the inner side strain of the specimen was generally greater than the outer side strain under the combined effect of constrained torsion and distortion.

(3) The developed fine finite element model simulated well the deformation characteristics and macrocurves of the three curved composite box beam specimens under a positive moment, as well as the normal strain distributions and trends of the concrete plate, steel webs, and steel bottom plate. The constrained torsion and distortion effects of each specimen were accurately predicted; that is, the finite element model results could accurately simulate the full-process force behavior of the three curved composite box beams. This finite element model can be used as a powerful numerical tool for subsequent parameter analysis of the force behavior of curved composite box beams.

(4) The ultimate bearing capacities of the specimens obtained from the test, the numerical model, and the theoretical formula were compared. There were good agreements among the results obtained from the test, the numerical model, and the theoretical formula. 

## Figures and Tables

**Figure 1 materials-14-02978-f001:**
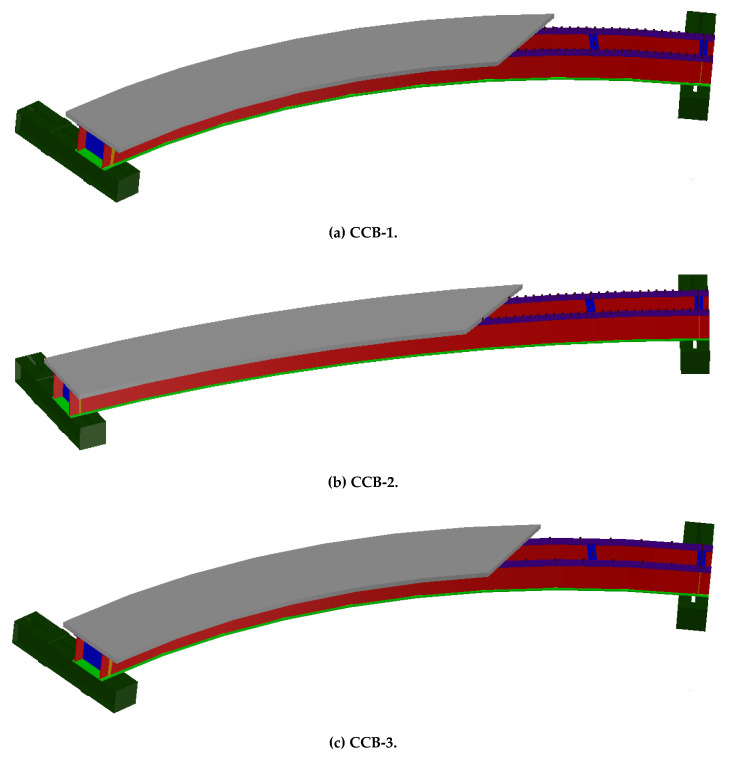
Schematic plots of the test specimens.

**Figure 2 materials-14-02978-f002:**
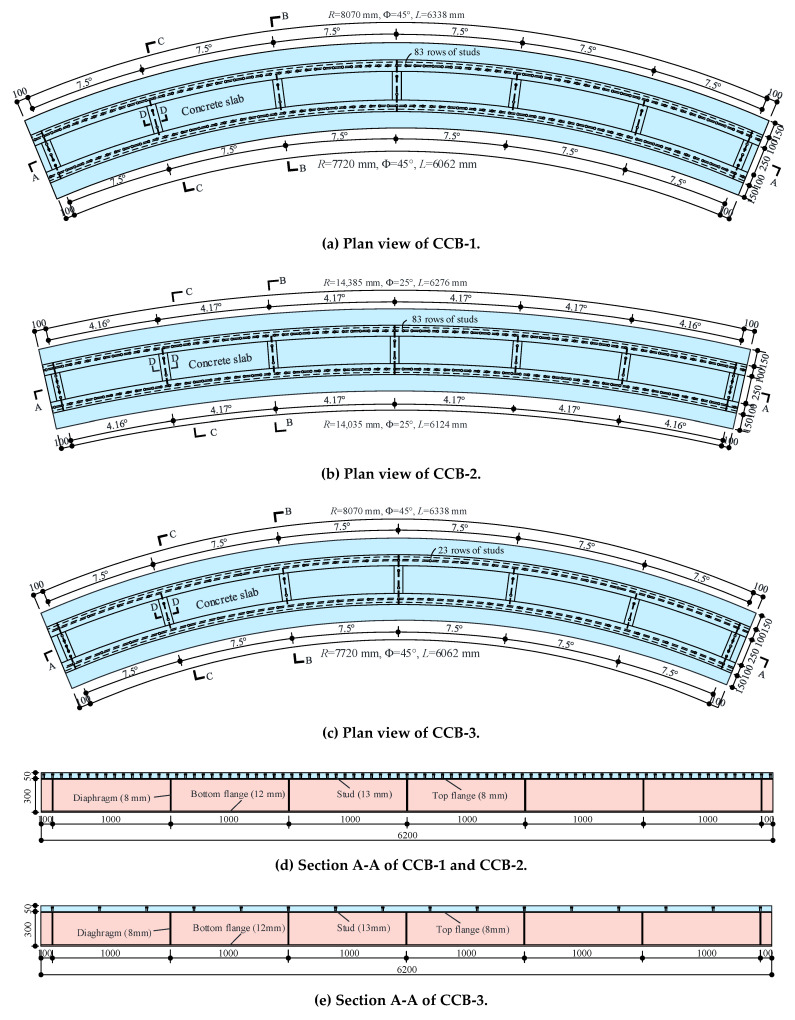
Details of the test specimens (in units of mm).

**Figure 3 materials-14-02978-f003:**
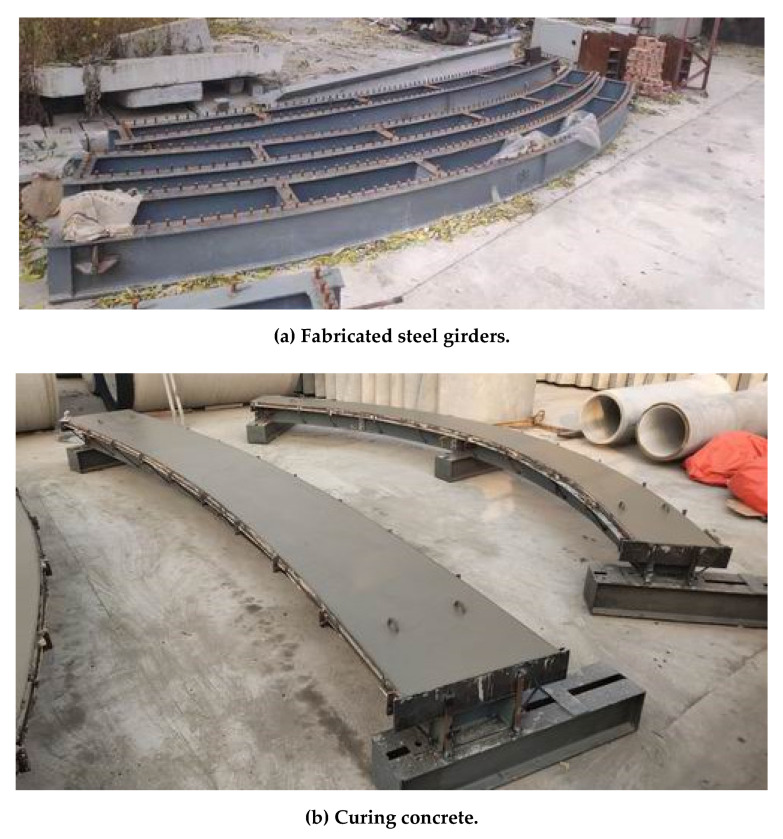
The manufacturing process of the test specimens.

**Figure 4 materials-14-02978-f004:**
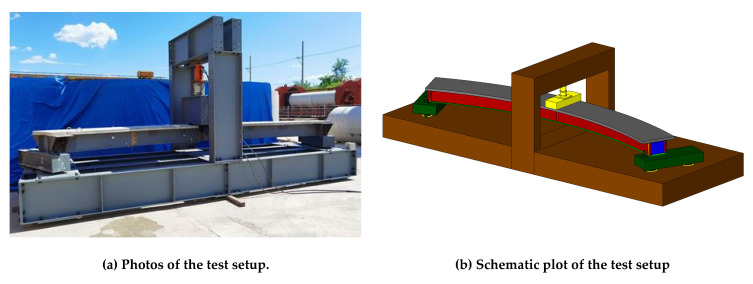
Photos and schematic plot of CCB-1.

**Figure 5 materials-14-02978-f005:**
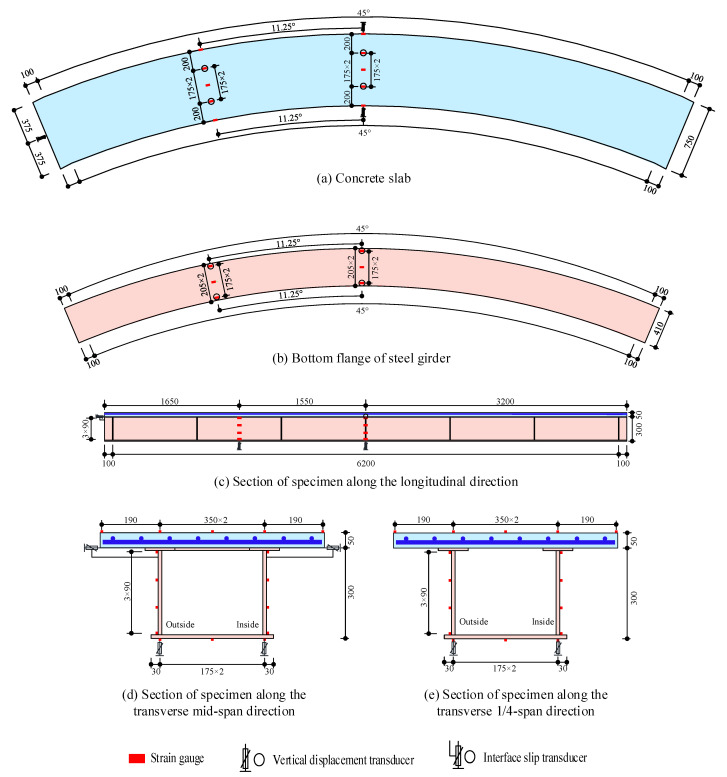
Measurement of CCB-1 (in units of mm).

**Figure 6 materials-14-02978-f006:**
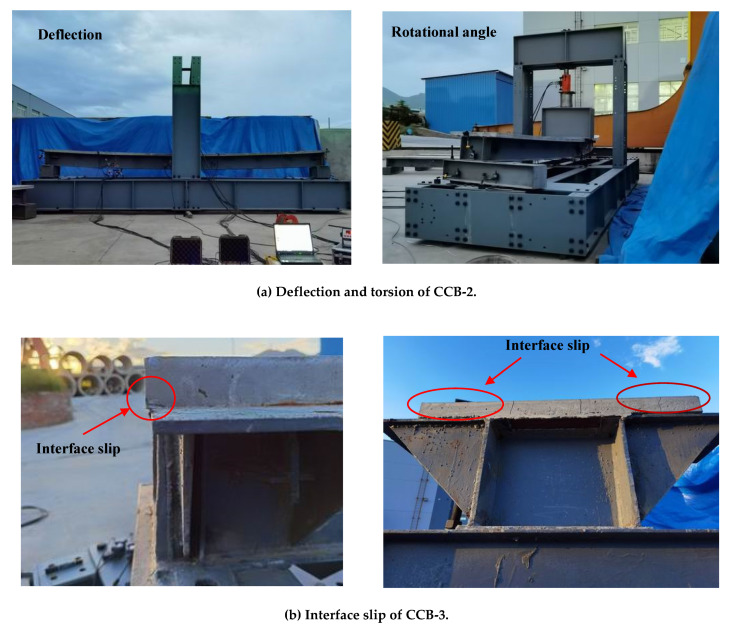
Test observations.

**Figure 7 materials-14-02978-f007:**
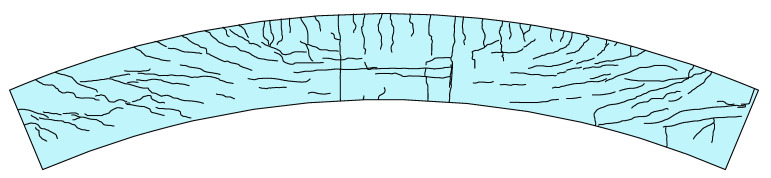
Crack profile of CCB-3.

**Figure 8 materials-14-02978-f008:**
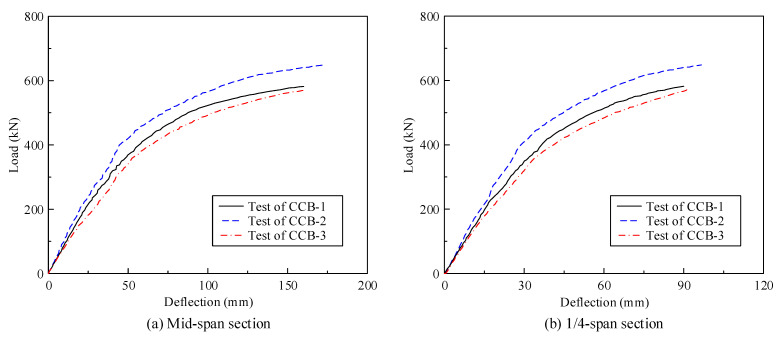
Load–deflection curves of the test specimens.

**Figure 9 materials-14-02978-f009:**
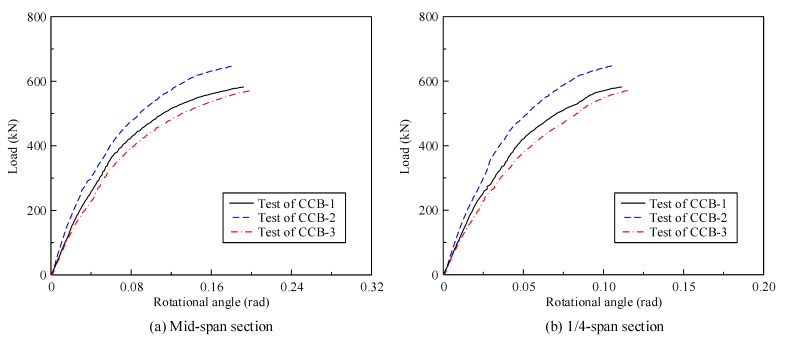
Load–rotational angle curves of the test specimens.

**Figure 10 materials-14-02978-f010:**
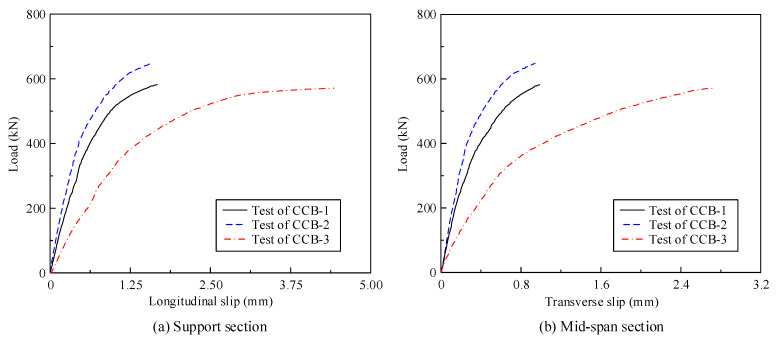
Load–interface slip curves of the test specimens.

**Figure 11 materials-14-02978-f011:**
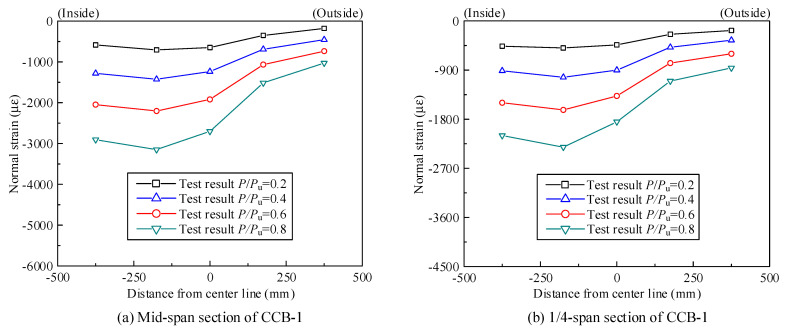
Normal strain distribution of the concrete slab or rebar of the test specimens.

**Figure 12 materials-14-02978-f012:**
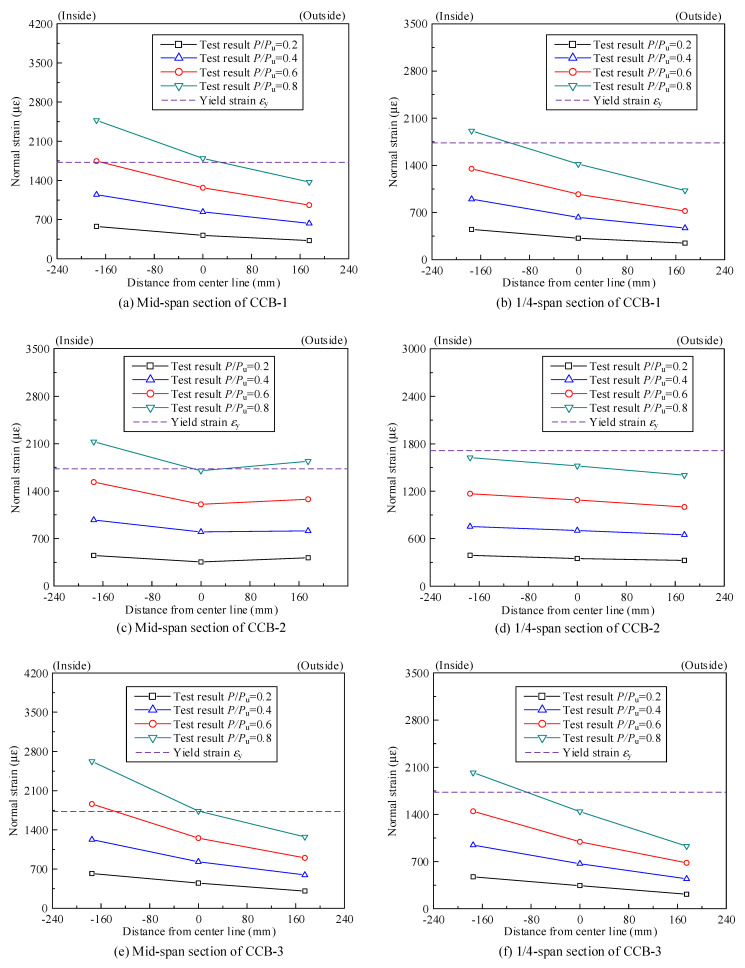
Normal strain distributions of the steel bottom flanges of the test specimens.

**Figure 13 materials-14-02978-f013:**
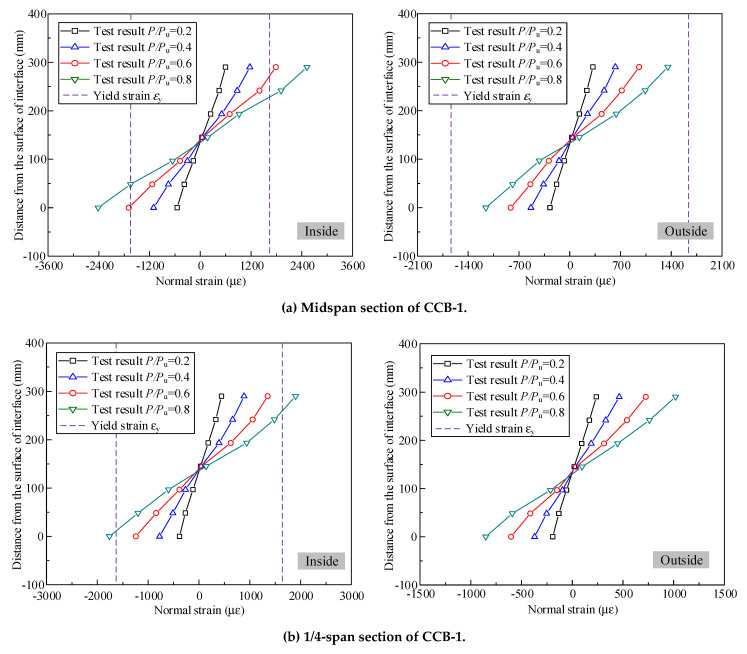
Normal strain distributions of the steel webs of the test specimens.

**Figure 14 materials-14-02978-f014:**
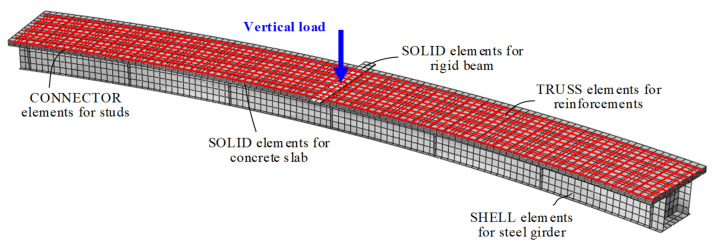
Finite element modeling scheme of CCB-2.

**Figure 15 materials-14-02978-f015:**
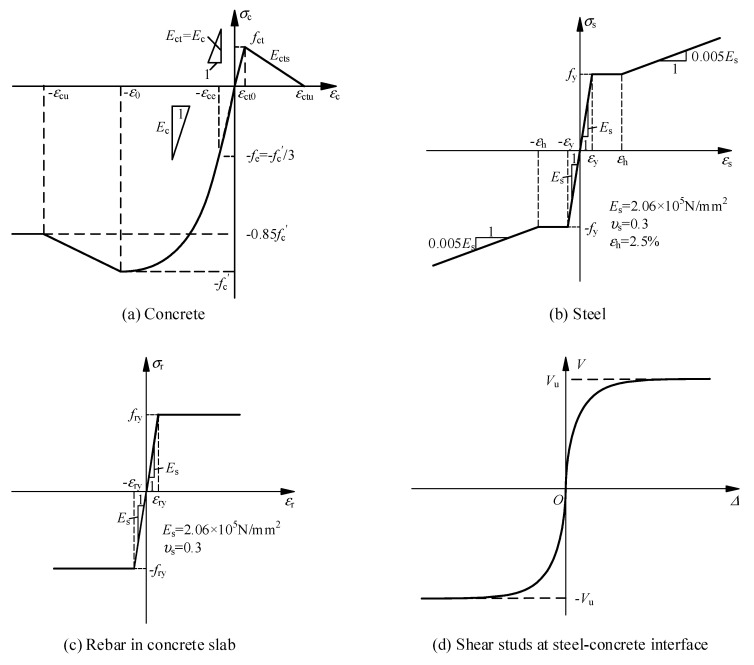
The uniaxial constitutive relationship of the FE model.

**Figure 16 materials-14-02978-f016:**
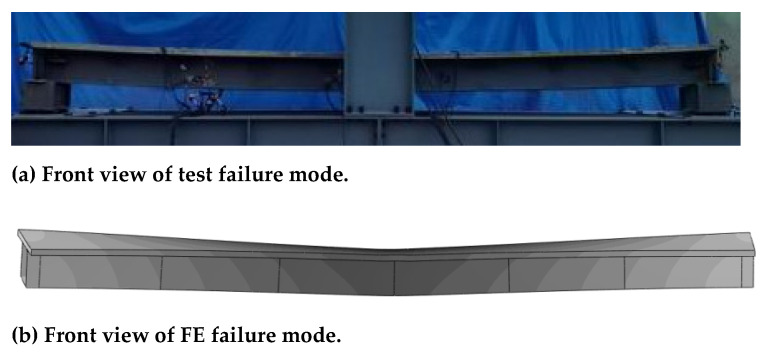
Comparison of the failure modes from the test and finite element analyses of CCB-2.

**Figure 17 materials-14-02978-f017:**
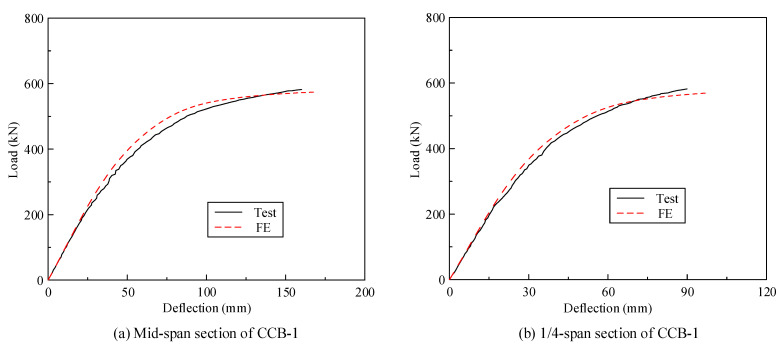
Comparison of the load–deflection curves from the test and numerical analyses.

**Figure 18 materials-14-02978-f018:**
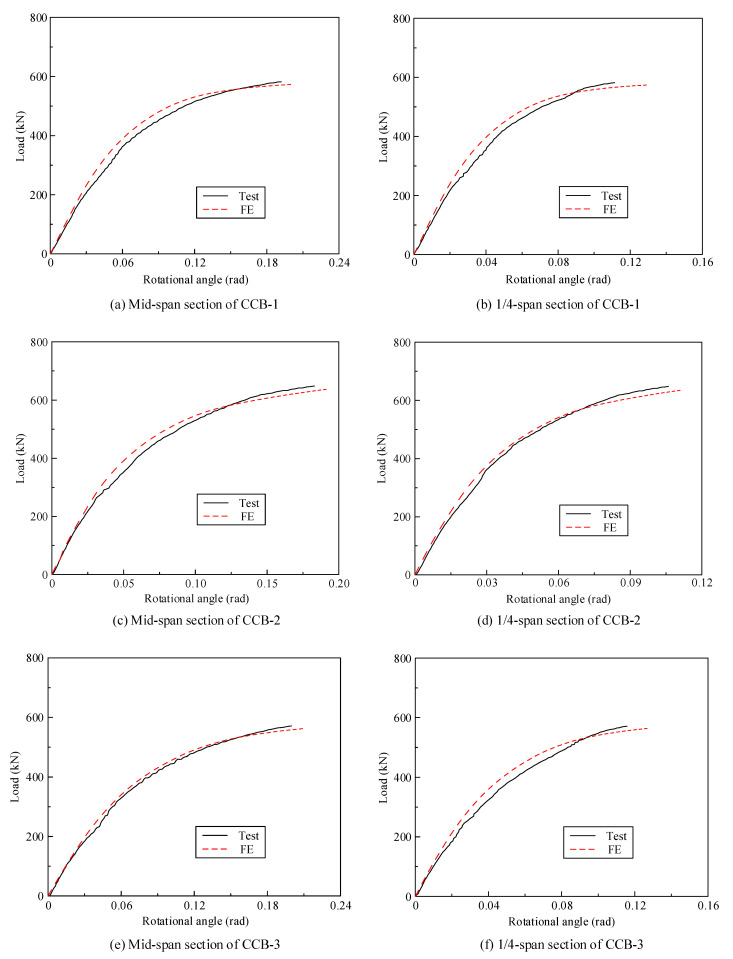
Comparison of the load–rotational angle curves from the test and numerical analyses.

**Figure 19 materials-14-02978-f019:**
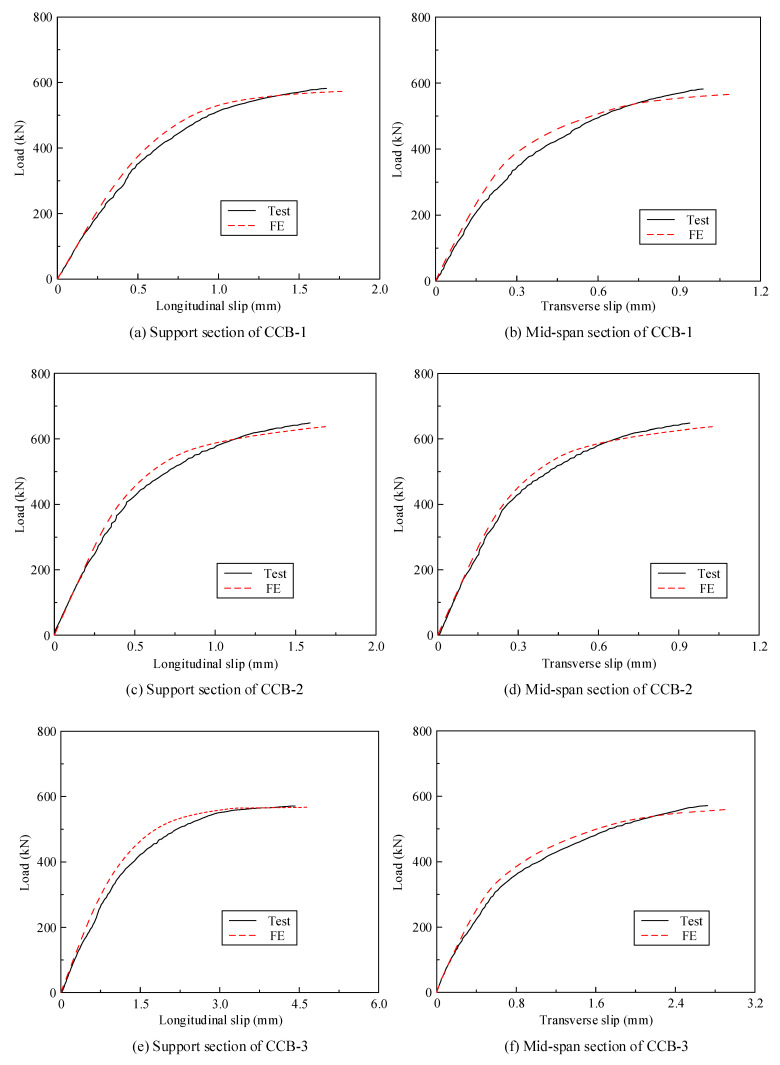
Comparison of the load–interface slip curves from the test and numerical analyses.

**Figure 20 materials-14-02978-f020:**
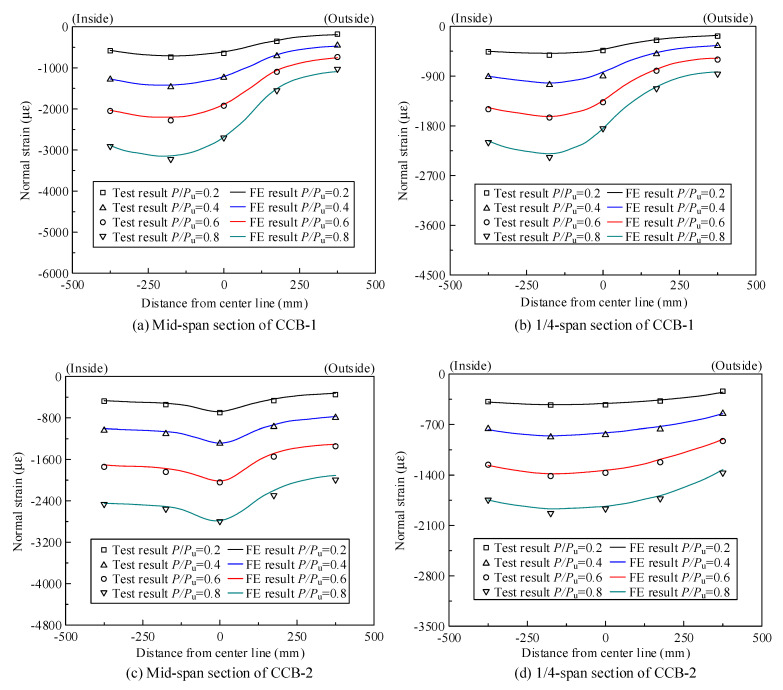
Comparison of the normal strains of the concrete slabs from the test and numerical analyses.

**Figure 21 materials-14-02978-f021:**
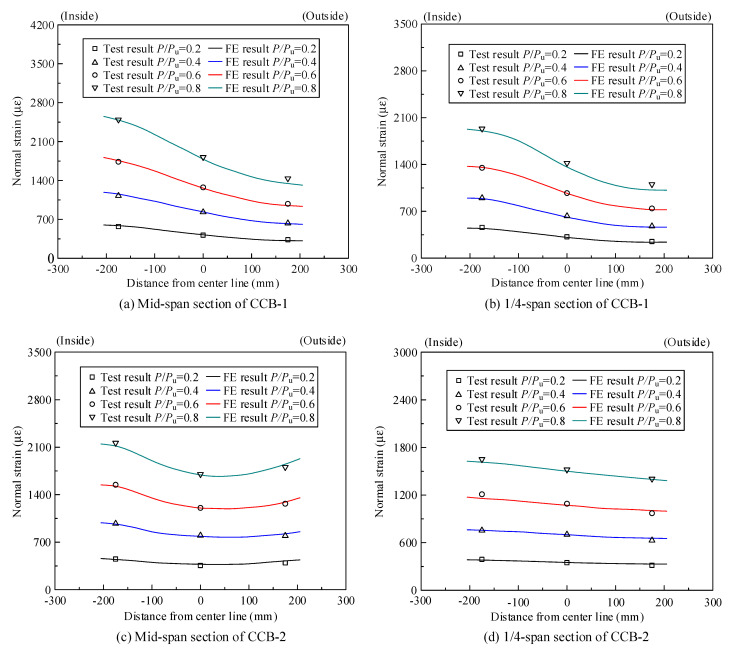
Comparison of the normal strains of the steel bottom flanges from the test and numerical analyses.

**Table 1 materials-14-02978-t001:** Parameters of the test specimens.

Specimen	Central Angle	Shear Connection Degree
CCB-1	45°	Strong
CCB-2	25°	Strong
CCB-3	45°	Weak

**Table 2 materials-14-02978-t002:** Material properties of the steel plates and rebar coupons.

Coupons		Q345c Steel (Thickness 8 mm)	Q345c (Thickness 12 mm)	HRB335 (Diameter 12 mm)
Yield strength (MPa)	Mean	350.1	352.5	337.4
	Standard deviation	4.3	3.4	4.8
Ultimate strength (Mpa)	Mean	570.7	560.5	496.0
	Standard deviation	6.7	7.1	5.8
Elongation ratio	Mean	0.304	0.316	0.324
	Standard deviation	0.0031	0.0027	0.0040

**Table 3 materials-14-02978-t003:** Mix proportion of C50 concrete coupons (in units of kg/m^3^).

Component	Water	Cement	Sand	Aggregate
Proportion	205	490	560	1197

**Table 4 materials-14-02978-t004:** Material properties of the C50 concrete coupons (in units of MPa).

Specimen		CCB-1	CCB-2	CCB-3
150 mm cubic compressive strength (MPa)	Mean	49.7	60.4	52.1
	Standard deviation	2.1	3.0	2.7

**Table 5 materials-14-02978-t005:** Yield load and capacity of test specimens.

Specimen	Yield Load (kN)	Yield Displacement (mm)	Ultimate Capacity *P*_u_ (kN)	Ultimate Displacement (mm)
CCB-1	491.8868	84.4649	581.9083	160.0512
CCB-2	526.3409	81.7638	648.2645	171.8867
CCB-3	459.3324	83.6047	571.2004	161.9660

**Table 6 materials-14-02978-t006:** Bearing capacities obtained from experimental, numerical, and theoretical results (unit: kN).

	CCB-1	CCB-2	CCB-3
Experimental	582	648	571
Numerical	574	638	564
Theoretical	570	630	570

## Data Availability

Not applicable.

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
