# Peer review of "Test and Numerical Model of Curved Steel–Concrete Composite Box Beams under Positive Moments"

_materials, 2021, doi:10.3390/ma14112978_

Round 1
Reviewer 1 Report
Reviewers' comments:
Manuscript ID: materials-1242905
Full Title: Experimental and numerical research on curved steel-concrete composite box beams under positive moments.
The manuscript describes Experimental and numerical research on curved steel-concrete composite box beams under positive moments. The manuscript needs a detailed editing. Some markings are made to just illustrate the extent of editing needed. A thorough revision addressing all the concerns is needed and if the authors are prepared to do that it can be considered for a review of the revised manuscript.
The authors need to consider the following comments
1) Some sentences need reconstruction and the level of English should be improved.
2) The introduction section should be improved; more related papers must be discussed and superiority, novelty, critical improvement in this study must be clarified.
3) Material properties - section should be detailed.
4) Please provides the references for all equations and formula.
5) Figures 2 and 7 – not clear make clear.
6) 3.3. Strain distribution - should be detailed.
7) Conclusions: the authors need to improve with more specific short results and conclusions.
8) Several faults: are added or missing spaces between words: see manuscript file.
9) References: there are recent references in 2020-2021 treating the same subject, you can use. And make all references in same format for volume number, page number and journal name.
So that I recommended this manuscript to major revision and for future process.
Author Response
Attached please find the coverletter file.

Reviewer 2 Report
The originality and the scientific value of the subject research are good.
The research area is experimental and numerical research on curved steel-concrete composite box beams.
The experimental program contains interesting experimental tests. However, the informative value needs to be improved.
The manuscript also includes numerical modelling, which is potentially interesting, but in its current form very insufficient.
It is necessary to significantly improve the entry into the issue and references.
Part of the reference is quite old [1], [2] and there is also a lot of reference in Chinese. It is necessary to prefer English.
References are not formatted according to the MDPI template.
E.g. Label of Table 3. is on the wrong side.
It is necessary to prepare the manuscript with greater interest.
The manuscript has the usual structure, but part of the discussion is missing!!!
Table 2. - also state the standard deviations or VoC.
Table 3. - also state the standard deviations or VoC.
Were the tests performed only for compressive strength?
Specify the concrete recipe (cement, aggregate, water, ..)
Figure 7. Is it possible to extend the image and the thickness of the cracks?
Clearly state the method and position of deformation measurement in the experiments.
Figure 16. - enlarge the pictures
Extensive research is underway in the area of nonlinear calculations of concrete structures when it is necessary to rework and expand the information in the introduction section.
These are mainly the possibilities of material models of concrete, approaches to the choice of parameters, or taking into account the uncertainties in the calculation or stochastic character of concrete.
Sucharda, O. et.al. Non-linear analysis of an RC beam without shear reinforcement with a sensitivity study of the material properties of concrete. Slovak J. Civil Eng. 2020, 28, 33–43.
Valikhani, A.et. al. Numerical Modelling of Concrete-to-UHPC Bond Strength. Materials 2020, 13, 1379
The chosen approach and procedure for numerical modelling is possible.
However, more information must be provided.
The overall similarity of the calculation and the informative value must be substantially improved.
It is necessary to provide detailed information about the computational model, parameters of the solver and boundary conditions.
It would be appropriate to take more account of the stochastic nature of the material properties of concrete.
Add input parameters for concrete, reinforcement, steel, interface, and solver in Tables.!!!
There is no clear procedure for determining the fracture mechanical parameters of concrete.
Overall, more calculations should be made to clarify the sensitivity (uncertainty) of the model and the effect of the input parameters for the concrete and its parts.
It is also necessary to improve the discussion of the results and to set limits on the calculations performed.
Overall, it is necessary to improve the informative value of the manuscript.
It is necessary to rework and expand also the introduction part (entry into the solved problem).
The manuscript must be revised before publication.
Author Response

(The authors gave the same response as above.)

Reviewer 3 Report
- the current study investigates the influence of positive moment acting on curved steal concrete composite beams. The authors carry experimental and numerical investigation under static loading. The authors found that the curved beam were significantly affected such that the curvature and interface shear connection degree affected the loading behaviour of the beam.
- Please consider revising the title and removing the word research, include the word modelling if possible.
- Please consider reviewing the abstract and highlight the novelty, major findings and conclusions.
- After line 112 please answer the following question: What is the research gap did you find from the previous researchers in your field? Mention it properly. It will improve the strength of the article.
- Consider renaming section 2 to Materials and Method (recommended)
- Table 2 it is not clear if the authors measured these parameters on their own or got it from somewhere else, if the later then the table needs to be referenced
- Please consider changing section 3 to results and discussion
- Line 248 “only a few cracks” this is very vague sentence, does that mean it is a significant damage or not? Please use more scientific wording to describe your experimental results
- Line 268-269 what about other designs does the shear connection degree have an influence on the bearing capacity? Please discuss and support with references
- Please consider combining figure 8-10 in one figure
- The authors should add a list of nomenclature for all the symbols and Greek letters reported in the manuscript
- There are way too many figures in the manuscript the authors should consider moving some to an appendix (there are more than 70 graphs!)
- What are the limitations in your FE model please mention some if any
- The results are merely described and is limited to comparing the experimental observation. The authors are encouraged to include a discussion section and critically discuss the observations from this investigation with existing literature.
Author Response

(The authors gave the same response as above.)

Round 2
Reviewer 1 Report
Reviewers' comments:
The authors revised the manuscript according to the reviewers' comments.
So that I recommended this manuscript accept for publication in Materials.
Reviewer 2 Report
The changes made the improvement of the manuscript.
The research area and results are from the context of the manuscript can better understand.
The manuscript presents an interesting topic, which combines the field of materials engineering - mechanical properties of concrete, structural tests of steel-concrete composite box beams, and the use of advanced numerical modeling.
The results of the research and information value of the manuscript can be evaluated overall well.
MDPI template is not used.
The manuscript can be published in the journal.
Reviewer 3 Report
All questions answered